# Food Waste of Italian Families: Proportion in Quantity and Monetary Value of Food Purchases

**DOI:** 10.3390/foods10081920

**Published:** 2021-08-19

**Authors:** Vittoria Aureli, Maria Luisa Scalvedi, Laura Rossi

**Affiliations:** CREA Council for Agricultural Research and Economics—Research Centre for Food and Nutrition, Via Ardeatina 546, 00178 Rome, Italy; marialuisa.scalvedi@crea.gov.it (M.L.S.); laura.rossi@crea.gov.it (L.R.)

**Keywords:** food waste, monetary value, food categories, household food waste, Italy

## Abstract

Halving per capita food waste (FW) is one of the objectives of the Sustainable Development Goals. This study aims to evaluate the weight and monetary values of food waste among a sample of Italian families. In a representative sample of 1142 families, the adults responsible for food purchases and in charge of preparing meals were assessed with a self-administrated questionnaire measuring quantity and typology of FW. These data were linked with food purchases figures measured as an average of four weeks. Italian families wasted 399 kg of food per week (4.4% of the weight of food purchased), which correspond to a monetary value of €1.052 (3.8% of the overall food expenditure). Clustering the food groups according to waste quantity, typology, and monetary value made it possible to show that price has a role in the generation of food waste, as the lower the unitary cost, the higher the quantity of waste. Consequently, foods with high unitary costs were less wasted. The results of this study showed that Italian consumers are sensitive to the economic impact of waste and this should be considered in sensitization campaigns.

## 1. Introduction

Reducing food waste (FW) along the entire food supply chain (FSC) is an important policy priority included in the UN Sustainable Development Goals (SDGs) for 2030 [1]. In fact, in the framework of the adoption of sustainable production and consumption models, Goal 12.3 recommends halving food losses and waste. In response to the SDGs, in 2018, the European Union encouraged the Member States to monitor and put in place actions aimed to reduce FW at every stage of the supply chain [2].

According to Willett et al. [3], reduction of food losses and waste is a key factor to increase the sustainability of the agri-food sector alongside optimization of the production processes and adoption of a healthy and sustainable diet. Food loss and waste was estimated to be one-third of the edible food as reported by the World Resource Institute [4]. These figures are equivalent to 1.3 billion tons of food produced for human consumption wasted each year [5] with an economic loss of €800 billion [6]. The geographical area and level of development of countries have a differential influence on where food is wasted along with the FSC. Households and catering sectors are the most impactful (53%) in high-income regions, while in low-income areas, food losses are the consequences of non-efficient products’ management and storage [7].

In this context, consumers have an important role considering that the food choices and behaviors have a clear impact on domestic waste production. As highlighted by Schanes et al. [8], families showed ambivalent attitudes towards waste prevention, between good intentions to waste reduction and personal preferences regarding food safety, taste, and freshness. As shown by Rohm et al. [9], a factor influencing consumers’ food waste is the refusal of products with imperfect physical appearance still proper for human consumption such as crooked cucumber, broken biscuits, or products in deformed packages or products that have a best-before date which is approaching or passed, but that are still perfectly fine to eat. Besides this, several socio-demographic determinants affected household food waste (age of children in the households [10], gender, education [11], income [12], composition and number of members of the household as well as culinary and buying food habits [13]). Moreover, the economic implications of food waste are considered more relevant for people than social or environmental consequences [8].

A study carried out by Philippidis et al. [14], that used a system-wide modeling approach to simulate the prospective economic impacts of reducing household FW by 2030, showed that a 50% reduction in household food waste would lead to a per capita saving of €93 for EU families. In the report of the Waste and Resources Action Programme (WRAP), the amount of avoidable FW generated each year by an average British family is approximately 210 kg, corresponding to €565.7 [15]. Data on FW domestic economic value are limited and collected with non-comparable methodologies. In an average Finnish household, FW accounted for €70/person/year according to Katajajuuriuri et al. [16]. Von Massow et al. [17] reported an FW monetary value of €15.31/family/week in Canada. A Tunisian study based on self-respondent questionnaires showed that 42.7% of participants realized they lost more than €5.10/month for domestic FW [18]. A study conducted among South Korean households assessed the cost of disposal and any other costs associated with FW quantifying in €2.90 the economic value of daily domestic FW for each family [19]. According to Notarfonso et al. [20], the economic value of food waste and loss along with the whole FSC in Italy is about €13 billion a year, with an average of 149 kg of food wasted per person.

In 2018 a first nationwide and country-representative measurement of Households’ Food Waste in Italy (HFWI) was carried out to quantify the food wasted off, to identify the food categories mostly wasted, and to evaluate in which conditions foods were thrown away. Results of this survey are reported in Scalvedi & Rossi [21]; in brief, for each family, the average quantity of food waste was 370 g per week with perishable products, such as fresh fruit and vegetables, bread, and milk that were most discarded. Considering the growing interest in socio-economic aspects of food waste [22], the team of the Italian Observatory of Food Surpluses, Recovery and Waste, in 2019, carried out a further study to evaluate the amount of food waste in weight and monetary values concerning the overall quantities of food purchased and the money spent for food commodities by the Italian families. Results of this assessment are presented and discussed in this paper that was conceived to answer the following research questions: (i) how much food is wasted compared to what is purchased in an Italian household? (ii) How much is the burden of the monetary value of the food wasted off on food expenditure of Italian families? (iii) Concerning waste and monetary value of commodities, is it possible to distinguish and characterize them based on the different food categories?

## 2. Materials and Methods

### 2.1. Design of the Study

In July 2018, a survey was carried out on a representative sample of 1142 Italian families extracted from a consumer panel of Growth for Knowledge (GFK) Italy^®^, a market research agency. According to Scalvedi & Rossi [21], sampling was carried out taking into account the geographical area, age, gender, income, and the number of people in the family and was based on National Institute of Statistics data. A sample of 4000 households was selected across the country and 2936 of them resulted as eligible. Finally, 1142 households completed the survey with a responding rate of eligibility accounting for 38.9%. The final sample of the household included in the study was in line with national data, mostly in terms of distribution per region and household size. Differences were observed as far as concerning small cities households that were 26% in our sample and 32% at the national level. Most respondents were female (61%), mostly 35–49 years old (42%), and with a medium education level (54%) [21]. Adults (over 18 years old) mainly responsible for food purchase and in charge of preparing meals were assessed with a self-administrated questionnaire for quantification of household food waste. The questionnaire was developed in a European context [23] and used in Italy [21] to have data comparable with other European Countries. The questionnaire is based on a recall methodology that is known to underreport the absolute quantities in comparison to diaries or direct analysis of waste compositional analysis [24]. For these reasons strategies to reduce the underestimation of the amount of food wasted off were put in place, e.g., asking people to pay attention to the level of waste in the week of the survey. On the other hand, to avoid bias related to socially desirable behaviour such as waste reduction reporting, respondents were asked not to change their usual attitude. The questionnaire [23] is reported in the Appendix A (Appendix A: The food waste questionnaires developed by van Geffen et al., 2017). To accomplish the objective of the present paper and to answer the research questions raised above, in this study, we linked and duly elaborated the data collected in the HFWI survey with the data that GFK systematically tracks on the consumer panel in terms of food products purchased in supermarkets and large-scale retailers either as the amount of food bought and recording its economic value.

### 2.2. Measures

HFWI data referred to week 29 (16–23 July) of 2018. Participants reported the amount of food wasted considering only the edible fraction of food intended to be eaten by humans excluding inedible fractions, such as bones, peels, seeds, stumps, etc. Closed-ended questions concerning 24 food groups were asked to measure food waste with practical units (i.e., tablespoons of vegetables, units of fruit, slices of bread, etc.) further converted in weight using the methodology developed by van Geffen et al. [23]. The conversion table is reported in the Appendix A (Appendix A: Conversion table for calculation of the food waste measure, in gram developed by van Geffen et al., 2017). For preparation such as pasta with tomatoes, we asked respondents to refer to the main ingredient that in this case is “pasta”. The same procedure was applied for ready meals. For example, in the case of frozen spinach with mozzarella, the category was “Non-fresh vegetables”. For each foodstuff thrown away, also the wastage state was asked, allowing one to categorize FW according to the following 4 typologies: (i) completely unused food (e.g., unopened packages, one apple, whole loaf, a milk package, etc.), (ii) partially used food (e.g., egg white thrown away after yolk usage, half a cereal package, half a milk package, etc.), (iii) meal leftovers (e.g., a spoon of beans, or pasta or rice left in the plate, the beverage that is left in the glass, etc.), (iv) leftovers after storing (e.g., fruit salad after it was stored, smashed potato after it was stored, beverage leftovers that are disposed of after these were stored, etc.). FW amount and proportion of different typologies as resulting from HFWI survey were processed to be coupled with food purchased, and its economic value. Specifically, the purchases’ weekly average and the corresponding economic values registered in the weeks 26–29/2018 (25 June–23 July) were estimated. It was decided to cover four weeks to include the purchase of fresh products and long-lasting commodities. In addition, it was considered that this timeframe is used also for Italian household food expenditure statistics [25]. Another methodological issue faced was the matching between HFWI food categories and GFK Consumer Panel food groups’ purchases resulting in 20 food groups analyzed in this study (Table 1). Four HFWI categories such as meat substitutes (e.g., vegetarian burgher) or soups, marginal in the Italian market, were excluded since they were not comparable with the GFK classification.

### 2.3. Data Analysis

#### 2.3.1. Development of Indicators of Food Waste Ratio on Food Purchases in Weight and Monetary Value

For each food category, the conversion factors defined by van Geffen et al. [23] were applied to transform the categorical answers in weight (grams) (Appendix A). The total amount of food wasted resulted as the sum of the estimated food waste of each category. The amount of food waste per state was calculated by dividing the amount of food waste of each category by the number of states ticked by the respondent. This value was reported as a percentage of the total food waste for each category. For example, if a respondent reported 100 g of vegetable food waste and ticked the unused and leftovers box, then we assumed that the respondent wasted 50 g of unused (50%) and 50 g of leftovers (50%) vegetable food waste [23]. Summarizing the percentages for each household and for each category, the total proportion (%) of typologies of food waste conditions for the 20 food groups analysed was estimated.

Two indicators of food waste ratio on food purchases were developed calculating the percentage (%) of waste in quantity (kg) and monetary value (€) concerning the overall amount of food bought by the assessed families (Figure 1).

The first indicator measures food waste proportion on the amount of food bought as the ratio (%) of the weight of food waste (kg) generated in the week of the HFWI survey to the weekly average amount of food (kg) purchased in the 4 weeks considered. This procedure of calculation was carried out for the total quantity of food waste and each of the categories separately.

The second indicator developed for this study was the proportion of the monetary value of food waste on the overall food expenditure of the assessed families calculated as the ratio (%) of the cost of food wasted off (€) assessed in HFWI to the weekly food expenditure obtained as the average prices of purchased products tracked by the GFK Consumer Panel system. The proportion of monetary value for food expenditure was calculated for each food category.

#### 2.3.2. Segmentation of Food Groups according to FW Quantity and Monetary Value

To address the research question of characterizing food categories in terms of waste and economic relevance, a multivariate analysis was carried out to classify the food groups according to the weight of discarded products, the monetary value of disposed of food, and the wastage status of commodities. Specifically, principal component analysis (PCA) was applied as a data reduction technique to summarize information and build synthetic indicators as the combination of the original variables [26]. PCA was applied on the typology of waste measured as a percentage of the wasted weight by unused, partially used, leftover, and stored leftovers. Besides, it was applied on food waste weight (kg) and unitary cost (€) of waste per kg. Finally, cluster analysis has been performed allowing us to group similar food categories into clusters and, at the same time, characterizing the profiles of the resulting clusters. Towards this aim, a two-stage procedure using 2 clustering techniques was applied to the principal components identified [27]. In the first stage, the hierarchical method of Ward, based on the Euclidean distance, has been chosen to identify the optimal number of groups; in the second stage, the non-hierarchical algorithm k-means was applied to impose the optimal number of groups found in the previous stage. Statistical analysis was carried out using IBM SPSS Statistics, version 25.

## 3. Results

### 3.1. Food Waste Proportion of Food Purchased in Italy

As shown in Figure 2, in the week of the survey, the assessed families wasted off 399 kg of food, corresponding to 4.4% of the weight of food purchased. In terms of cost, the total weekly monetary value of food wasted off was €1.052 corresponding to 3.8% of the food expenditure of the sample.

Among the 20 food groups analysed, perishable foods such as yogurt, fresh fruit, and potatoes were wasted off mostly unused; non-alcoholic beverages, sweets, cereals for breakfast, and snacks were wasted off mainly after partial use; while pasta and rice have been thrown away mainly as leftovers (Figure 3).

In Figure 4, the food waste ratio on food purchases in weight and monetary value in the different food groups is reported. Potatoes (including preparations such as puree) (26%) and bread (19.2%) resulted in the highest food waste rate expressed in weight. Other categories of fresh products such as fresh vegetables (6.8%), and fresh fruit (4.8%) showed a relatively low incidence of food waste weight. Other categories that showed relevant figures of these indicators were “preserved fruit” (10.3%), “rice and other cereals” (9.4%), and “yogurt and other milk-based snacks” (9.3%). The ratio of waste in monetary value calculated since the average cost of different food categories showed a similar pattern observed for weight with “potatoes (including preparations such as puree)” (23.9%) and “bread” (19.7%) covering the largest quota of the economic value of disposing of food, followed by “yogurt and other milk-based snacks” (9.4%), “rice and other cereals” (8.1%), and preserved fruit 7.4%. The monetary values of waste of “fresh vegetables” and “fresh fruits” correspond, respectively, to 7.1% and 4.8% of the overall expenditure for these products of the assessed families.

### 3.2. Clusters Analysis of Food Groups

The first PCA, applied to percentages of the typologies of food waste, provided the first component as a synthetic indicator distinguishing two clear polarities of food groups: unused foods and partly used foods versus leftovers and stored leftovers (data not shown). The explained variance of that component accounted for 45% of the total variance of the original dataset. The second PCA provided a component accounting for 72% of the total variance of the original dataset that well represented the weight–price dualism with the negative correlation (−0.5) between weekly weight and unitary cost of the food waste off. Cluster analysis applied to the first components found in both PCAs provided three clusters (Figure 5): (i) wasted unused products at a high monetary value (Cluster 1) including 9 food categories, (ii) leftover and stored leftover products (Cluster 2) including 4 food categories, and iii) wasted unused products in high volume (Cluster 3) including 7 food categories. The analysis demonstrated that price has a role in the generation of food waste, as the lower the unit cost is, the higher is the quantity of waste that is generated within food groups. In other terms, foods with a high unitary cost are less wasted off (Cluster 1) and foods with low unitary cost are thrown away in large quantities (Cluster 3). The waste that comes from the foods included in cluster 2 is less influenced by the price, being essentially related to their use in the kitchen as leftovers. Perishable products are present both in Cluster 1 (e.g., meat, fish, etc.) and Cluster 2 (fresh fruit and vegetables, bread, etc.).

Further analysis of the three Clusters provided interesting information on food waste typologies across food groups (Table 2). Cluster 1 has a low proportion (11%) in the weight of food wasted off and a high proportion in the economic value (28%) of FW. Cluster 2, which includes food discharged as leftovers—stored and not stored—has small proportions of FW either in weight or economic value. On the other hand, Cluster 3 included the highest percentage in weight and monetary value of the food wasted off. In terms of unitary cost, Cluster 1 shows a higher mean food monetary value (€6.70) compared to the values found in the other Clusters (Cluster 2: 1.80€; Cluster 3: €2.20). Cluster 3 included foods that have been purchased in large quantity (5.831 kg/week) while Cluster 1 items had the highest purchase value (€16.569), even though unused wasted products bought in large quantities (Cluster 3) are those that have the greatest impact in both aspects considered, as a proportion of weight (5.5% vs. Cluster 1: 1.7%; Cluster 2: 4.7%) and monetary value (7.1% vs. Cluster 1: 1.8%; Cluster 2: 4.6%).

## 4. Discussion

The results presented in this article contributed to adding information on the impact of food waste generated at the very end of the food supply chain, at the consumer level. According to our data, Italian families discharged 3.8% of the weight of the food purchased, losing 4.4% of the economic value of the money spent for food. We decided to focus on the assessment of the economic value of food waste at the households, considering the importance of this aspect in the general debate on the evaluation of money wasted for producing food that is not used for human consumption. According to Vittuari et al. [28], food waste has a hidden burden because it can be considered as a “double waste” of energy: metabolic energy that is lost for food that is not eaten, and energy for food production that is lost when food is discharged. In addition to that, according to von Massow et al. [17], avoidable household food waste is a substantial contributor to global heating for the effect of greenhouse gas emissions, inefficient agricultural land use, and water loss. As mentioned, halving per capita food waste is one of the objectives fixed by the United Nations among the Sustainable Development Goals. Thus, the quantification of all the phenomena related to food waste is of key importance for monitoring purposes, especially in a context such as Italy, in which data on the monetary value of food waste is still anecdotal and collected at the local level often with estimation, not with measures, and published as grey literature [29,30].

In the light of these considerations, the absolute figures provided on weight and monetary values of food wasted off by the Italian families are relevant seeing the general lack of this information in Italy, to the best of our knowledge. However, the added monetary value of this paper is the analysis of the different food groups and the different effects on the waste of products about their prices and respect to the quantities that were normally bought. According to present data, in Italy, waste is mainly unused or partially used and in general, all the food that is cooked is consumed, leading to a generally small proportion of leftovers. Among the discarded unused food, there is a polarization related to price and weight, with food with high unitary cost that impacts less in the weight of food waste (Cluster 1) and food with low unitary cost that is thrown away in large quantities (Cluster 3). This is regardless of the perishability of the items; in fact, fresh foods are present both in Cluster 1 (e.g., meat, fish) and in Cluster 3 (e.g., fresh fruit and vegetables). This is in line with the general attitude of Italians to pay more attention to the issue of food waste essentially for ethical and economic reasons [31,32,33]. The economic crisis of the last decades could explain this finding as reported in a survey carried out by the National Confederation of Farmers [34] in 2011, in which the observed reduction of food waste was correlated with economic difficulties of the families. Among the measures taken to reduce food waste, it was reported there was an increased attitude to buying wisely (47% of respondents), a reduction of the quantity of food purchased (31%), increased use of leftover for other meals (24%), and the most attention to expiration dates (18%).

The issue of personal abilities as a means to reduce food waste is reported also by Janssens et al. [35], in a study showing that consumers did not have sufficient attention at the early stages of spending management, but they can compensate with correct knowledge of food conservation practices resulting in a small number of products thrown away, especially for the most perishable commodities (meat, fish, fresh fruit, etc.).

There is a consensus on the fact that the perishability of products is a determinant of food waste. In a review of several EU national studies, De Laurentiis et al. [36] showed that a proportion ranging from 44% to 47% of household FW is attributable to fresh products (fruit and vegetables). The same results were reported by Katajajuuriuri et al. [16], showing that vegetables and potatoes, fruits and berries, meat, fish, and ready-to-use foods such as pizza, burgers, etc., are wasted off more than long-lasting items. Even during the COVID-19 pandemic, fruits and vegetables were the foods most discarded by families of different nationalities according to the study of Filho et al. [37]. In addition, our data of the HFWI confirmed this point [21], which is in line with data coming from other European Countries [23,38,39]. The added information of the present study is related to the fact that among the perishable products, the most wasted off are those with the lowest unitary prices that are also bought in the highest quantities.

Shopping purchases of individuals have a huge impact on the level of FW, as reported by Fanelli [40] in an exploratory online survey carried out in Italy showing that the amount of food that is thrown away decreases when individuals buy foods in small markets and with a weekly frequency of two times/week or more. In our survey, only purchases done in large retail chains have been recorded by the GFK Consumer Panel, missing the food purchases of local markets or small stores. Thus, probably, what we observed in Cluster 3 is related to the behaviour of buying large quantities of food at low unit costs for the entire week that led to the increase in throwing away large quantities of perishable products.

The three clusters resulting in our analysis are characterized by a gradient in the amount of waste, with Cluster 1 having lower quantities of food waste than Cluster 2 and 3 and the waste quantity gradient is a direct consequence of the unitary prices of the items included in the 3 groups. Another aspect that could explain the differences observed in the three clusters is related to the Italian food habits and frequencies of consumption [41,42]. Most of the foods included in Cluster 3 are consumed daily or more than twice a day such as fruits, vegetables, bread, non-alcoholic beverages (that include milk), and yoghurts. On the other hand, Cluster 3 included foods that are consumed every week such as meat, fish, cheese among basic foods, and candies, crisps, nuts, and alcoholic beverages among comfort foods. This is another interesting polarisation of food groups that could be explained by the different attitudes of planning food purchases. We could speculate that large quantities of foods that are consumed every day and several times a day are more likely to be bought in incorrect quantities than foods that are consumed less frequently. Modalities and frequencies of consumption could explain the food items that the procedure of analysis included in Cluster 2. Pasta, rice, vegetable were wasted off mainly as leftovers, being long-lasting products, so unlikely to be wasted unused, and consumed as ingredients in recipes’ preparations. Waste as leftovers is less common in Italy than other types of waste, as confirmation of the fact that Italians tend to eat all the foods they cook [43].

Overall, in our sample, 399 kg of foods in one week were wasted off out of the 9.157 kg of foods purchased, with differences across the food categories. The high weight of foods wasted indicates that food waste substantially affects the overall quantities of food purchased. This may occur if the consumer is not fully aware of the correct storage practices, such as for “Potatoes and Preparations” or “Bread”, the categories with the highest proportion of weight on waste. Similarly, for “Yogurt, pudding, and fresh snacks”, waste resulting from a limited knowledge of how and/or for how long the product can be stored. “Rice or cereals” are wasted more than “Pasta”, even if they are purchased in smaller quantities, and this may depend not only on a lack of storage information but also on insufficient creativity in the use of these products once cooked. As mentioned, these items are also most wasted off as leftovers.

The comparison of the food waste ratio indicators—weight and economic value—on food purchases across food groups (Figure 4) showed that when the rate in weight is greater than that in monetary value, this corresponds to an average cost of foods wasted off lower than the average price of the category as it is for “Potatoes and potatoes-based food”, “non-fresh fruit”, “Rice”, “Breakfast cereals”, “Pasta”, and “Fish”. On the contrary, as happens for “Bread” and a few other products, when the proportion of waste in value is higher than that in weight, the wasted quantity has an average price higher than the average purchase price of the food’s items of the category. Similar percentages corresponded to similar prices of food wasted off and food purchased.

The overall value of FW found in this study was €1.052 out of €27.611 of the weekly food expenditure of the assessed families. This led to a value of waste of €0.92 per family per week. A value that is quite far from those found in the literature that reported, €15.31 [18] €5.10 [19], €2.90 [20], and €70 [17]. There are objective difficulties of comparisons of crude data of the monetary value of FW collected with various methodologies and in different contexts in which the cost of living and the impact of the cost of foods on family income is very different. At the national level, our data are lower than those reported by Waste Watcher [44] in 2021, which claimed that Italians throw away 529.2 g of food per week per person, considering what is left on the plate, in the fridge, and kitchen store cupboard. In 2020, the same group reported that Italians wasted €4.90 weekly per household for a total of about €6.5 billion, a decrease of 25% compared to the monitoring in previous years [29].

Our figures on the value of food waste are related to the food expenditure as registered by GFK on the consumer panel corresponding to €27.611 per week of the assessed families (1.142), corresponding to a monthly amount of €97 per household. However, as reported by the National Institute of Statistics, in Italy, the food expenditure accounted for €462 per month for each family [25]. This means that our data underestimate the measurement. Several reasons could explain this underestimation: differences among food groups of the two assessments and the fact that data registered by GFK are related to large retailers’ supermarkets, without considering other food purchases sources such as local shops and neighbourhood markets. In addition to that, the National Institute of Statistics survey [25] has a different methodology with a large sample size continuously monitored across the year while our assessment is a timely data collection limited to four weeks in 2018. The underestimation of overall food expenditure consequently determined an underestimation of the economic value of food waste. This is the most important limitation of this study. Data on food waste collected with recall questionnaire underreported the absolute quantities in comparison to diaries or direct analysis of waste compositional analysis [24]. The mechanism of underreporting of food waste with the use of a recall questionnaire is similar to those found in the studies on food consumption in which people recorded less food than they eat, underestimating consumption by a considerable margin, which in Italy was estimated at 29% [41]. Despite underestimation, the national food consumption data were considered a reference for the population. Although direct measurements of food waste produce more accurate results, these require most of the expertise, time, and cost. Direct measurements would be more appropriate in research settings than for monitoring. On the other hand, with the same approach used for the establishment of a nutritional surveillance system [45], it is possible to use data on food waste at the population level also with methods that underestimate the phenomenon, clarifying the purposes of data collection. This will allow one to estimate the extent of the problem, to bring together the data to identify specific groups, to provide information that allows the development of corrective actions, and, through continuous monitoring, to evaluate the effectiveness of the interventions themselves. The authors considered that the data reported in this paper fit with the idea of surveillance as described. Even though the absolute figures reported of the household’s monetary value of food waste have the mentioned limitations, to the best of our knowledge, this is the first attempt at its quantification at the national level with data representative of the Italian population.

## 5. Conclusions

The economic consequences of household food waste are a complex issue, with a multitude of causes that impact food purchases. There is an objective difficulty of the evaluation of net numbers because the methodologies of data collection are expensive and demanding in terms of capacity and resources. The findings from our research suggest the existence of differences in food groups characteristics that influence household food waste generation, with foods of high economic value that are less wasted compared to foods with low unitary costs. On the other hand, low unit-cost food is purchased in large quantities. This leads to a higher accumulation of waste that has an impact on both quantities and absolute economic costs. Thus, strategies aimed at reducing food waste should take such differences into account when promoting behavioural and dietary changes.

The situation that emerged from the analysis of the data of the present paper, together with the interpretation of data of the 2018 survey, indicates an influence of purchases in the generation of FW at the household level. The lack of purchasing planning increases the quantities of FW and its impact on the food expenditure that, however, could be compensated for by consumers’ skills and knowledge about proper food preservation practices [8,35].

Concerning the economic aspects, the results of our study showed that the proportion of the value of waste is lower or similar to the proportion of waste expressed as quantities. In Italy, when there is an increase of quantities of food purchased, as experienced by Italian families during the 2020 COVID-19 lockdown, this does not lead to an increase in waste, but rather to a reduction [46,47,48]. The attitude of Italians is to eat all the foods that are bought and cooked, in particular, high price items. Present data demonstrated that price has a role in the generation of food waste, with foods’ categories with high unitary costs that were less wasted and foods’ categories with low unitary cost being thrown away in large quantities. Italian consumers are very sensitive to the economic impact of waste and this should be considered as a key message in sensitization campaigns. As mentioned, the limitation of this study is related to the underestimation of food waste with the use of recall methodology. Future research issues should be aimed to develop methods of correction of the measurements, e.g., developing a coefficient of comparisons among diaries and questionnaires to convert the different assessments and facilitate comparisons.

## Figures and Tables

**Figure 1 foods-10-01920-f001:**
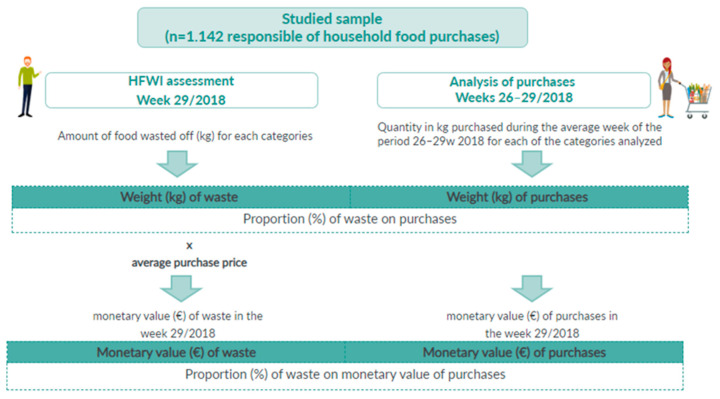
Methodology of the development of Indicators: proportion (%) of food wasted off in weight (kg) and monetary value (€) on family food purchases.

**Figure 2 foods-10-01920-f002:**
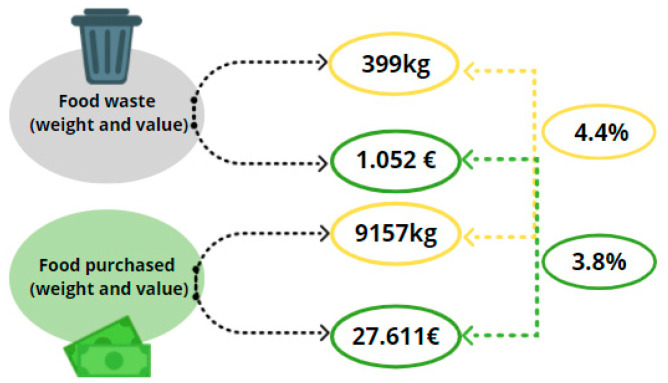
The proportion of food waste (%) on weight (kg) and monetary value (€) of food purchased by Italian families in the week of the survey.

**Figure 3 foods-10-01920-f003:**
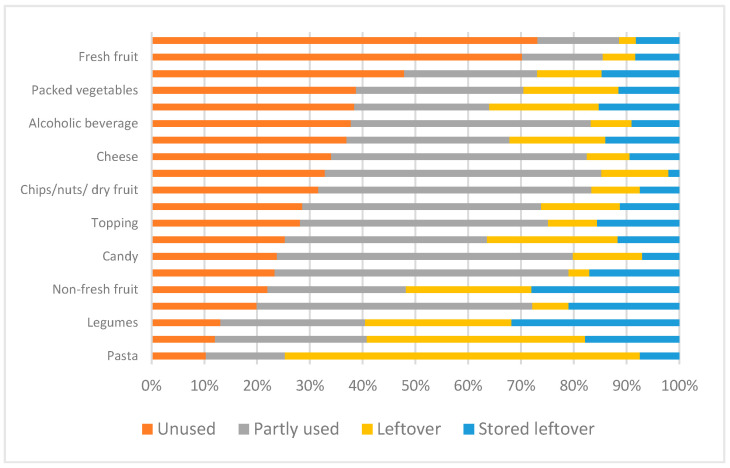
Proportion (%) of typologies of food waste conditions for the 20 food groups analysed.

**Figure 4 foods-10-01920-f004:**
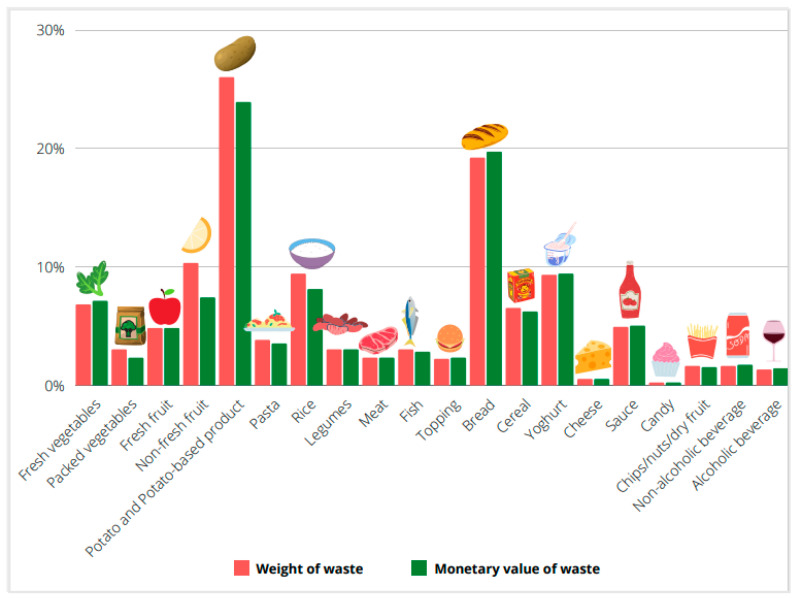
Food waste ratio (%) on food purchases related to weight and monetary value in the different food groups.

**Figure 5 foods-10-01920-f005:**
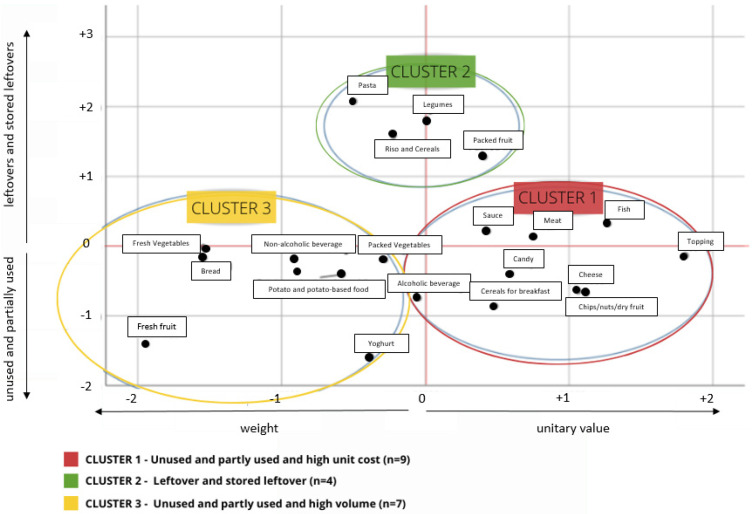
Cluster analysis diagram: clustering of the 20 food categories by the typology of waste and the weight-price indicator.

**Table 1 foods-10-01920-t001:** The 20 food groups were studied after matching HFWI food categories and GFK Consumer Panel food groups’ purchases.

HFWI Food Categories	Consumer Panel Food Groups’ Purchases
Fresh vegetables (except potatoes)	Fresh vegetables (except potatoes)
Processed Vegetables (canned/jar/frozen)	Pickled vegetables, in oil vegetables, frozen vegetables, tomato puree, and concentrate, peeled and tomato pulp, olives
Fresh fruit	Fresh Fruit (except dried fruit)
Processed Fruit (canned/jar/frozen/dehydrated)	Preserved fruit, fruit spirits, dehydrated, dried, frozen fruit
Potatoes and potato preparations (e.g., puree, pre-cooked potatoes)	Potatoes, puree, frozen potatoes, pre-cooked potatoes
Pasta	Dry and fresh pasta, ready-to-use refrigerated pasta (lasagne, cannelloni, etc.)
Rice or cereals for the preparation of first courses (e.g., spelled, couscous, etc.)	Rice, spelled, barley, couscous, other cereals, rice-based or cereal-based ready-to-eat dishes (frozen or from the refrigerated counter)
Legumes (e.g., beans, chickpeas, lentils, etc.)	Dried legumes, preserved legumes (chickpeas, beans, broad beans, lentils, peas, mixed)
Meat (excluding cold cuts used for the sandwiches)	Meat imposed weight (frozen, sausages, precooked), meat variable weight, canned meat
Fish	Preserved fish (tuna, salmon, sardines, etc. canned), fresh fish weight imposed and variable, frozen, fish specialties (smoked salmon, tuna carpaccio, shrimp, etc.)
Sandwich fillings (e.g., cold cuts, sliced cheese, cream spreads, etc.)	Cured meats, salami, cold cuts, sliced cheeses, spreads, salted pâté, and spreads
Bread	Fresh bread imposed and variable weight, industrial bread and sandwiches
Breakfast cereals (e.g., muesli, oatmeal, puffed rice.)	Breakfast cereals
Yogurt, puddings, fresh fruit snacks, etc.	Yogurt, fresh desserts, fresh snacks, cakes (puddings, panna cotta), sweet cheeses
Cheese (e.g., seasoned, fresh, grated, excluding sliced sandwich cheese)	Cheeses, weight-imposed, and variable (excluding sliced cheese)
Sauces/condiments (e.g., ketchup, mayonnaise, etc.)	Fresh seasonings (butter, margarine), sauces and pâté, mayonnaise, vegetable cold sauces/spreads, béchamel sauce, cream, fresh ready-made sauces.
Sweets (e.g., snack cakes/biscuits/chocolate/candy, etc.)	Cookies, sweet snacks, pastries, chocolate bars, chewing gum
Crisps/peanuts/nuts	Nuts, savoury after-meal (chips, savoury snacks)
Soft drinks (e.g., milk, fruit juice, carbonated drinks, excluding water, tea, coffee, syrups)	Fresh drinks, carbonated drinks, fresh and long-life fruit juices, soft drinks and non-alcoholic beers, fresh and long-life milk, milk-based drinks
Alcoholic drinks	Wine, alcoholic beers, alcoholic aperitifs, liqueurs, champagne/sparkling wine

**Table 2 foods-10-01920-t002:** Characteristics of the three clusters: weight and value of food waste and food purchased resulting from the cluster analysis.

Clusters	Categories	Wasted Volume (kg/Week)		Wasted Monetary Value (€/Week)		Wasted Monetary Value Per kg	% Waste in Weight	% Waste in Monetary Value	Weight of Food Purchased	Monetary Value of Food Purchased	% Purchases in Weight	% Purchases in Monetary Value
	N	kg	%	€	%	€	%	%	kg	€	%	%
Cluster 1. Wasted unused products at high monetary value	9	45	11%	300	28%	6.7	1.7%	1.8%	2628	16,569	29%	60%
Cluster 2. Leftover and stored leftover products	4	33	8%	60	6%	1.8	4.7%	4.6%	698	1322	8%	5%
Cluster 3. Wasted unused products in high volume	7	321	81%	692	66%	2.2	5.5%	7.1%	5831	9720	63%	35%
Total	20	398	100%	1052	100%	2.6	4.4%	3.8%	91,567	27,611	100%	100%

## Data Availability

The archived data and all the elaboration and analysis generated and used for the presentation of results in this study are fully available on request from the corresponding author.

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
