# Peer review of "Food Waste of Italian Families: Proportion in Quantity and Monetary Value of Food Purchases"

_foods, 2021, doi:10.3390/foods10081920_

Round 1
Reviewer 1 Report
Comments are written in the attached file.

Author Response
We thank the reviewer for his/her comments and suggestions. We addressed all the aspects that needed improvement. We refer the corrections to the line numbers, figures and tables of the revised manuscript. Corrections are reported in track change.

Reviewer 2 Report
Overall comments
This paper has interesting findings in terms of the typologies of food wasted and estimation of the cost of waste in Italy which as the authors point out will inform more specifically consumer campaigns about reducing food waste. I am not an expert in PCA, therefore, I will not be commenting on the details of this analysis. However, there are some changes that need to be made to the paper to improve the clarity and flow of the text. While I have made some corrections below I suggest a review of tense, use of plurals and general grammar across the whole paper. Use of the word values could perhaps be replaced with cost or monetary value to improve the understanding of what you are presenting.
I am also not particularly familiar with Euros however I found the use of decimal points/fullstops and commas when Euros were quoted not consistent throughout the paper.
The quality of the grammar also needs to be reviewed for clarity and flow of the paper.
I was not familiar with use of the term 'wasted off' and would prefer 'waste' and use 'wasted', or discarded where relevant.
Abstract
I am unsure of the meaning of the word 'values' in the second sentence. Perhaps use the term 'monetary value' or cost (please review the use of this word in the entire paper e.g. L239, and Table 2 headings.
L12-13 This sentence needs rewording, perhaps is should be worded '....FW in weight and monetary value (or use the word 'cost') with respect to the overall .....' or simplify the aim to 'This study aims to evaluate the weight and cost of food waste among a sample of Italian families.
L15-16 Please add in a time period e.g. per week to inform readers.
L18-19 'made it possible....' (or remove so it states '...monetary value showed that price...') and 'as the lower the unitary cost, the higher the ...'
L15 the sentence 'These data were gathered...' perhaps replace gathered with 'linked.'
L20-21 This sentence 'Results of our study showed that Italian consumers pay attention to the price of products they buy.' This makes it sound like you asked consumers this question. I think the abstract should end with 'The results of this study showed that Italian consumers are sensitive to the economic impact of waste and this....'
Rather than the first sentence as I think this overstates the results.
Introduction L32-34 This sentence needs rewording. Do you mean that one-third of the food produced for human consumption is wasted? e.g. food losses and waste was estimated to be one-third of the food produced for human consumption
L36-38 This sentence could be reworded with '...the geographical area and level of development of countries has a differential influence on where food is wasted along the FSC'.
L33-34 Remove 's' from production
L41 '....consumers have an important role....'
L47 What is buckle?
L63 The dollar symbol needs to be first
L64 remove s from the word 'disposals'
L79 the first aim needs to be reworded for clarity. Similarly, the third aim also needs rewording for clarity.
L86 change to 1142 Italian families
L96 Isn't this '...and recording its ....'
Measures
Please indicate how many categories were excluded.
Figure 2 use decimal points in percentages, i.e. 4.4 rather than 4,4%. This is also the case with other figures where a comma is used instead of a decimal point. How have you calculated 3.8%? When I calculate it, I get 2.62%?
L182-184 It is stated that 1052 is a global figure; however, perhaps the use of the word 'total' is more apt as 'global' may be confused with a figure for the entire world rather than from your study. (I was unclear when I first read this).
L223 This sentence needs some rewording. Suggest replace 'less' with 'lower' and 'large' with 'higher.' Also not sure of the use of the word 'thought' in this sentence.
L226-228 This sentence needs rewording.
Table 2
Title - replace 'value' with 'monetary value' or 'cost'
Suggest table headings are reworded to include that columns are means.
Suggest reordering the columns to cluster the columns related to purchasing and waste together. Currently, there is a mix of columns which make it harder for the reader to follow.
For consistency, add '%' to the final two columns to make them consistent with the '% waste in weight' and the '% waste in value' columns. (Note previous suggested change from value to cost.
Discussion
L263 check spelling of anecdotal
In the first paragraph, I suggest you make a small addition about the contribution of food waste to Greenhouse gas emissions.
L276-278 This sentence needs rewording.
L283 Does this sentence need the word 'most'
L285 suggest the addition of the word 'to' to this sentence e.g 'as a means to reduce...'
L309-310 Check tense of these sentences e.g. change thrown to throwing
L316 Change large part to 'Most of the foods...' or 'a large proportion'
L317 Add-in 'and yoghurts'
L323 Is there a reference for this sentence?
L327 Suggest there is an 's' added to leftover. There is a mistake in this sentence 'o'? should this be 'of'? also think that 'typology' needs to be changed to 'typologies' as I think you are referring to more than one typology.
L329 Is that in one week, if so please include.
L330 Suggest changing the word 'discharged' to 'wasted'
L340 Are these ratio indicators for individual food categories available?
L349 I am not very familiar with Euros but I wonder if using commas is more useful for numbers over 1000 otherwise it may be perceived as a decimal point but the journal may have guidelines about this. Is this 1052 Euros?
L356 Perhaps say how they differ i.e. lower or higher
L357 states 529,2 g, should this be 5,292g?
L359 Is that 49 or 4.9 Euros? Need to be consistent with the use of commas
L362-364 My calculation shows 108, not 97 therefore unsure how this is being calculated. I suggest the number in brackets is represented as (1,142) as it refers to the sample size.
L372 Unsure of the meaning of 'punctual - do you mean opportunistic or other meaning?
L382 add an 'if' e.g. 'even if it is well known...'
L383 change 'most' to 'the most' or 'more'
Sentence L381-385 is long and hard to follow
L389 'the possibility of ....'
L389 Confusing sentence
L393 'to our best knowledge'
L402 - 403 '...foods of high economic value are less wasted compared to (or with respect to) foods with low unitary cost.
L403-405 Reword to improve understanding.
L407-409 I am unclear why you are saying that your study influenced the planning of purchases.
L413-415 This sentence needs rewording to improve clarity.
L415-L421 Similar to my comment about the end of the abstract I think your last sentence could be reworded to more accurately reflect your findings. There is little in the final paragraph about food categories which was one of your aims.
Author Response

(The authors gave the same response as above.)
